# Evaluation of Wound Healing Activity of Salvianolic Acid B on In Vitro Experimental Model

**DOI:** 10.3390/ijms22147728

**Published:** 2021-07-20

**Authors:** Urszula Szwedowicz, Anna Szewczyk, Krzysztof Gołąb, Anna Choromańska

**Affiliations:** 1Department of Molecular and Cellular Biology, Wroclaw Medical University, 50-556 Wroclaw, Poland; a.szewczyk@umed.wroc.pl; 2Department of Animal Developmental Biology, Institute of Experimental Biology, University of Wroclaw, 50-328 Wroclaw, Poland; 3Department of Pharmaceutical Biochemistry, Wroclaw Medical University, ul. Borowska 211A, 50-556 Wroclaw, Poland; krzysztof.golab@umed.wroc.pl

**Keywords:** wound healing, salvianolic acid B, cell migration, collagen, Danshen

## Abstract

Despite a wide range of bactericides and antiseptics, the treatment of chronic or complicated wounds is still a major challenge for modern medicine. Topical medications are the most sought-after new agents for use as treatment. The therapeutic concentration of their active substances is easy to achieve with the lowest possible burden on the patient’s body. This study assesses the effect of salvianolic acid B (Sal B) on the proliferation, migration, and production of collagen type III by fibroblasts, which are the most important processes in wound healing. The study was conducted on human gingival fibroblasts obtained from primary cell culture. The results showed that Sal B at a dose of 75 µg/mL increases the cell viability with significant stimulation of the cell migration as demonstrated in the wound healing assay, as well as an increase in the expression of collagen type III, which has great importance in the initial stages of wound scarring. The results obtained in the conducted studies and previous scientific reports on the antibacterial properties and low toxicity of Sal B indicate its high potential in wound healing.

## 1. Introduction

Despite many advances over the century in antibacterial pharmacotherapy and wound healing, still, several types of wounds are difficult to treat and nurture. In many chronic medical conditions such as diabetes, vascular diseases, or immunodeficiencies seemingly harmless injuries can quickly turn into complicated wounds [1,2]. The pathophysiology of these diseases disrupts the already complex wound healing process which slows down at the stage of inflammation and the formation of granulation tissue [3]. The duration of bacterial infection has a crucial impact on the following stages and quality of scar tissue [4]. The presence of exudate and its components has a degrading effect on proteins, extracellular matrix, and growth factors make wound closure impossible [5]. The main goal of wound care is to promote wound healing in the shortest possible time with minimal side effects. It is also important for the tissue to regain its former features without abnormal wound repair, e.g., to hypertrophic scars [6]. Accurate diagnosis with an individual approach to the patient’s medical background and removing obstacles to healing is the most important in wound bed preparation. Without comprehensive treatment affecting every phase of wound healing, it is not possible to succeed even with advanced therapies [7]. Antiseptics, while essential, are insufficient and toxic to newly formed tissue or even disturb its physiological remodeling [8]. For this reason, many experts advise against using them on open wounds for a long time [9]. Still, it sought a middle ground by which we obtain a promising broad antimicrobial spectrum in action with non-toxic effect to mammalian cells. With this in mind, it is worth looking at substances that are already known in clinical practice, but, for example, in a different field.

A well-known and commonly used substance in Chinese, Korean and Japanese medicine is an extract derived from the dried root of Salvia miltiorrhiza, a widely used plant in Traditional Chinese Medicine, which contains various active compounds [10]. One of them is salvianolic acid B (Sal B), a water-soluble phenolic acid with antioxidant properties, as previously reviewed [11]. Sal B is used as the treatment in cardiovascular diseases. It inhibits ischemia and enhances angiogenesis in vitro [12,13]. Other pharmacological activities of Sal B include inhibition of platelet aggregation and its role in vascular tone regulation [14,15]. It is usually given in the form of Fufang Danshen Tablet or Fufang Danshen Dripping Pill to patients who suffers from systemic diseases or for example angina pectoris and myocardial infarction [16,17,18]. Importantly, Sal B has anti-inflammation activity and the ability to inhibit the TGF-β signaling pathway [19]. Phenolic acids are also known for antibacterial properties and are not cytotoxic to human cells [20,21]. These features, among others, encouraged us to conduct preliminary research to establish whether there are any prospects for Sal B to be used in wound treatment. As Chen et al. (2014) proved, Sal B may enhance the proliferation of fibroblasts cells and have an impact on collagen biosynthesis [22]. We decided to study the potential of these properties, and also took a step further to see if this acid affects cell migration, an especially important part of wound closure and healing.

## 2. Results

### 2.1. Effect of Salvianolic Acid B on Cell Viability

Cell viability was measured calorimetrically using the MTT assay and the values of different absorbances were shown as a percentage of control. Fibroblasts treated with salvianolic acid B for 48 and 72 h showed that the survival rate increased slightly even with mildly toxic compounds the MTT assay may give slightly enhanced apparent viability due to stress-induced elevated metabolism. The observed effects may be ascribed to such effects as the effect seen appears not to be concentration-dependent, in all concentrations (Figure 1).

Extended incubation time increased cells’ viability, but the peak was at a lower concentration (75 µg/mL) compared to 48 h (100 µg/mL). In both cases, the increase was over 35 percentage points. However, the cells’ mitochondrial activities at 100 and 150 µg/mL are very similar in both times, suggesting that Sal B and its mentioned properties have a positive effect on fibroblasts’ growth at a low dose.

### 2.2. Wound Healing Assay

The assay was performed to assess whether salvianolic acid B affects the migration of fibroblasts. The cells were incubated in a medium without compound and with Salvianolic acid B at concentrations of 25, 50, and 75 µg/mL. The cells were analyzed after 24, 48, and 72 h with a microscope (20×). The use of Ibidi Culture Inserts made it possible to standardize the test and obtain reproducible results. The following photographs show the process of 500 µm cell free gap closure (Figure 2).

ImageJ made it possible to clearly visualize the process of space overgrowing with cells and at the same time to obtain numerical data. Figure 3 shows how fibroblasts grow in the certain concentration and incubation times.

Based on the data presented in Table 1, we can conclude that salvianolic acid B affects the growth and migration of fibroblasts and it is an interaction dependent on the concentration of substances in the cell’s environment. First, two concentrations of salvianolic acid B (25 and 50 µg/mL) demonstrated their most promising effects after 24 and 48 h of incubation. Compared to the control, the cells migrated faster, already covering 56.89% and 51.63% of the gap area after 24 h, while in the control it was only 20.41%. However, after 72 h, the percentage of occupied area was comparable for control cells and the other two concentrations (99.40%, 99.49%, 98.42%). A different effect of Sal B is shown at a concentration of 75 µg/mL. In this case, the compound reduces cell mobility, causing more than two times less surface occupation (42.09%) on the third day of incubation.

### 2.3. Immunofluorescence Assessment of Collagen Type III Expression Level

Indirect immunofluorescence was performed to evaluate the effect of salvianolic acid B on the production of type III collagen in fibroblasts. Three different concentrations of Sal B added to the culture medium were taken into consideration, i.e., 25, 50, 75 µg/mL. Cells without Sal B constituted the control to which the remaining samples were compared. Slides were analyzed under a confocal microscope; the obtained results are shown below in Figure 4.

In all samples with the tested substance, fluorescence was more intense in comparison to control. The highest collagen type III expression was observed in cells incubated with 75 salvianolic acid B (Figure 5), 1.5 times greater than in lower concentrations.

## 3. Discussion

It has already been proven that salvianolic acid B not only has antibacterial properties but also anti-adhesion activity against bacteria helpful in combating wound infection or even biofilm creation, which can lead to the sensitization of bacterial cells to any antiseptic or antibiotic [23]. Sal B itself has a synergistic or additive effect with commonly used antibiotics in the fight against methicillin-resistant Staphylococcus aureus (MRSA) strains [24]. The efficiency of the inflammatory phase also depends on the environment in which it takes place. While researching psoriasis-like dermatitis, it was found that Sal B increases skin hydration and a moist wound with low oxygen tension has optimal conditions to close quicker with an increased re-epithelization [25].

Many pre-clinical and clinical studies have reported that salvianolic acid B can stimulate the development of new blood vessels. As Lay et al. proved, Sal B enhances angiogenesis in vivo and inhibits skin flap necrosis while VEGF and VEGF-R2 gene expression increased [13,26]. This suggests that the described chemical compound may be able to improve the blood supply of newly formed scar tissue.

This current study was designed to determine if salvianolic acid B, a bioactive compound of *Salvia miltiorrhiza*, has the potential to become a useful agent in the treatment of complicated or chronic wounds. We took into consideration the most important features that enable the evaluation: effect on fibroblasts’ proliferation, migration, and collagen production. The MTT Assay confirmed that Sal B propagates fibroblasts to divide and grow depending on the concentration of the tested substance. The positive effect can already be seen at the lowest chosen concentration of 25 µg/mL in 48- and 72-h incubations. Cells developed particularly well at the highest concentrations used, i.e., 75, 100, and 150 µg/mL. These results show that even a small amount of salvianolic acid B has an influence on the fibroblasts and also confirms that the substance itself, used at a higher concentration, is not toxic.

The results of the Wound Healing Assay suggest salvianolic acid B accelerates gap closure and the most effective concentration is 25 µg/mL. The results obtained after 24 h emphasize the greatest difference in the effect of the tested concentrations of the Sal B. Rapid wound closure is a quick formation of a barrier against a hostile external environment, which may be the key to regenerative healing. However, the 75 µg/mL concentration inhibits this process almost completely. At this concentration, the results obtained in the Wound Healing Assay and the MTT Assay are not consistent. This shows that salvianolic acid B has a different optimal concentration range for various processes, e.g., proliferation, cell growth and migration. This leads to the conclusion that the compound may have an impact on extracellular matrix (ECM) and cell-cell interaction. For example, it can increase production and release of migration stimulating cytokines or other substances, like proteinases, which enable cells to detach from the bottom and fill the space. These processes can be impacted by Sal B itself and/or its products of metabolism.

Production of collagen type III is associated with early phases of wound healing and serves as scaffolding supporting subsequent processes. It is found in granulation tissue, in a larger amount than mature skin [27]. After promising results of MTT and Wound Healing assays, we decided to analyze the collagen type III production in human gingival fibroblasts with the use of indirect immunofluorescence. The analysis confirmed that salvianolic acid B stimulates collagen biosynthesis in all tested concentrations. Compared to the control, the highest collagen synthesis is observed in the sample containing the compound at 75 µg/mL. It is consistent with the obtained result of MTT assay. Lower doses of Sal B tested accelerated the migration of the fibroblasts, while higher doses tested (75 µg/mL) limited cell migration with a simultaneous increase in survival. This suggests that Sal B can influence the production and secretion of substances, i.e., growth factors, that affect the production of type III collagen. However, it has been proven that Sal B inhibits pathological fibrosis of the liver or myocardium inhibiting the synthesis of type III collagen. This suggests that the tested compound has a different effect on collagen production in physiological and pathophysiological processes [28].

All results presented in this work confirm the beneficial effects of salvianolic acid B on human givingal fibroblasts. Accelerated cell migration and increased collagen type III production show that Sal B affects many phases of wound healing which suggests the possibility of applying locally throughout convalescence. To get the full picture of the issue, it would be advisable to extend the research with tests on fibroblasts and their differentiation into myofibroblasts, the very important cells in the wound contraction and closure. A lot of information on this matter could be provided by measuring the expression of the smooth muscle-specific protein α-smooth muscle actin (α-SMA), the marker of myofibroblasts [29]. However, the skin does not only consist of fibroblasts but also keratinocytes, melanocytes, and mesenchymal stem cells which are cooperating in the process of wound healing. It would be worth investigating the effect of salvianolic acid B on them. Promising results would be a great introduction to in vivo experiments and clinical tests.

## 4. Materials and Methods

### 4.1. Materials

Salvianolic acid B (≥95%, analytical standard) was purchased from Sigma-Aldrich (St. Louis, MO, USA) in the form of dry powder, which was dissolved in purified water to obtain 5 mg/mL concentration. Human gingival fibroblasts were obtained from a patient. Dulbecco’s modified Eagle’s medium (DMEM), fetal bovine serum (FBS), and trypsin ethylenediaminetetraacetic acid (trypsin EDTA) were purchased from Sigma-Aldrich (St. Louis, MO, USA) and penicillin-streptomycin from Lonza (Basel, Switzerland). The compound 3-(4,5-dimethylthiazol-2-yl)-2,5-diphenyltetrazolium bromide (MTT) was purchased from Sigma-Aldrich (St. Louis, MO, USA).

### 4.2. Cell Culture and MTT Assay

The in vitro study was conducted on human gingival fibroblasts obtained from primary cell culture. The protocol of deriving a cell line from a tissue section was accepted by the Bioethics Commission of Wroclaw Medical University, No. KB-434/2017. The tissue cultures of human gingival fibroblasts were derived from healthy adult volunteers. The epithelial-connective tissue fragment was removed in a small portion (1–2 mm^2^) using the “punch” method. This method makes it possible to obtain the connective tissue layer located closest to the epithelium, which is characterized by effective keratosis [30].Fibroblasts were cultured in DMEM with needed additives in an incubator with 5% CO_2_ at 37 °C. The cells were counted and seeded in a 96-well plate at a density of 8 × 10^4^ cells per well. Fibroblasts were treated with various concentrations of salvianolic acid B for 48 and 72 h. After that time, cell viability was determined via an MTT assay after incubation with different concentrations of the salvianolic acid B (25 µg/mL, 50 µg/mL, 75 µg/mL, 100 µg/mL, 150 µg/mL). The MTT test assesses cells viability through assessing cell metabolic activity. The mitochondrial metabolic function was expressed as a percentage of viable treated cells in relation to untreated control cells. The absorbance at 560 nm was measured using GloMax^®^ Discover multimode microplate reader (Promega, Madison, WI, USA). 

### 4.3. Wound Healing Assay

Based on the MTT assay results, the salvianolic acid B concentrations were selected for the wound healing assay (25 µg/mL, 50 µg/mL and 75 µg/mL). For wound healing assay fibroblasts were plated into a 6-well plate with Ibidi Culture Inserts. After 24 h, when cells had created the monolayer, the inserts were removed. The wells were filled with DMEM and salvianolic acid B in a specific concentration, then incubated for 24, 48, and 72 h. Cell culture was observed under the CKX41 Olympus microscope (Tokyo, Japan). Software ImageJ (LOCI, University of Wisconsin) was used to prepare graphics of visualization of wound closure which was created on the basis of the results of the experiments.

### 4.4. Immunofluorescence Assessment of Collagen Type III Expression Level

Based on the MTT and wound healing assay results, the salvianolic acid B concentrations (25 µg/mL, 50 µg/mL and 75 µg/mL) and the incubation time (48 h) for confocal immunofluorescence study were selected. The confocal laser scanning microscope (CLSM, Olympus FluoView FV1000) was used to visualize collagen type III changing after incubation with different concentrations of salvianolic acid B. Cells were harvested on cover glasses in Petri dishes overnight. The medium was altered to medium with different extract concentration and incubated in 37 °C by 24 h. Afterwards, samples were washed in PBS and fixed in 4% of formaldehyde. Then cells were permeabilized with 1% Triton X-100 in PBS for 5 min, raised in PBS 3 × 5 min, blocked with 4% Bovine Serum Albumin (BSA) in PBS for 1 h and the polyclonal rabbit Anti-Collagen III antibodies (Abcam, Cambridge, MA, USA; ab7778; dilution rate: 1:100) for 3 h at 37 °C was added. PBS was used three times to wash samples and the secondary goat anti-mouse antibody, Alexa Fluor 488 (Invitrogen, A-11029), was added at a concentration of 1:100 for 1 h in 37 °C. Post incubation, the cells were washed three time in PBS, and the samples were mounted in Fluorescence mounting medium with DAPI (DAKO) to visualized nuclei. Images were taken using CLSM with oil immersion objective lenses in magnification 60x.

### 4.5. Statistical Analysis

The statistical analysis was performed using the GraphPad Prism 8 software (GraphPad, DMW Communication, San Diego, CA, USA). Values of *p* < 0.05 were considered significant. All results were expressed as mean ± SD. Differences between groups were assessed by two-way analysis of variance (ANOVA). The experiments were performed in three replicates.

## 5. Conclusions

The obtained results indicate that salvianolic acid B has a stimulating effect on human fibroblast cells. Its most promising property is the stimulation of cell migration, enabling the wound to close quickly. Simultaneous stimulation of cell migration and antibacterial action can significantly reduce wound inflammation and shorten its healing time. Moreover, the effect on cell migration and the production of collagen type III shows that salvianolic acid B affects both the initial and final stages of wound healing. This suggests the possibility of its topical application throughout the convalescence period. The presented research will be the basis for performing experiments on the animal model.

## Figures and Tables

**Figure 1 ijms-22-07728-f001:**
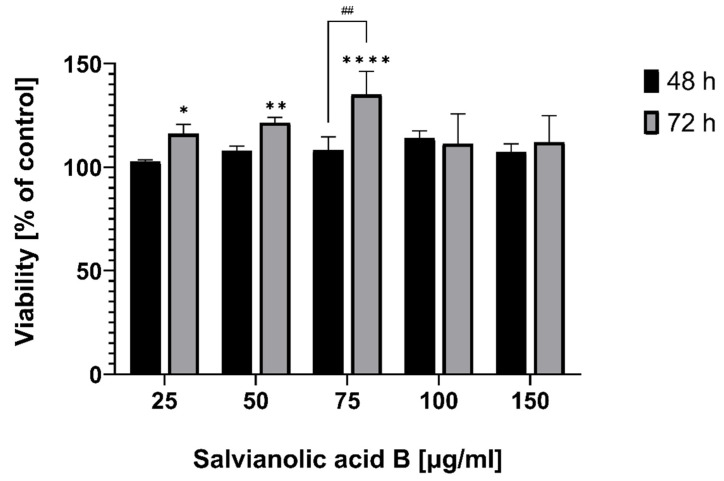
Human gingival fibroblasts viability measured by the MTT assay after 48 h and 72 h of incubation in different salvianolic acid B concentrations. Notes: (mean ± SD) *n* = 3, * *p* < 0.05, ** *p* < 0.01, **** *p* < 0.001 compared to control; ## *p* < 0.01 compared different incubation times.

**Figure 2 ijms-22-07728-f002:**
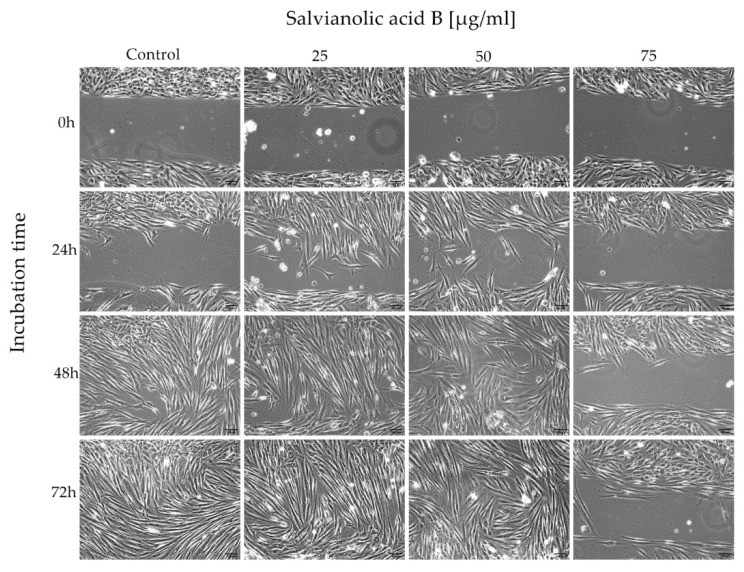
Human gingival fibroblasts after 24, 48 and 72 h of incubation.

**Figure 3 ijms-22-07728-f003:**
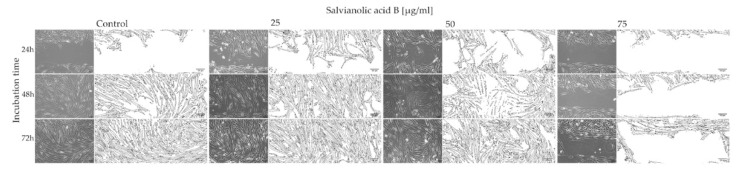
Microscopy studies and visualizations of salvianolic acid B-untreated and -treated fibroblasts.

**Figure 4 ijms-22-07728-f004:**
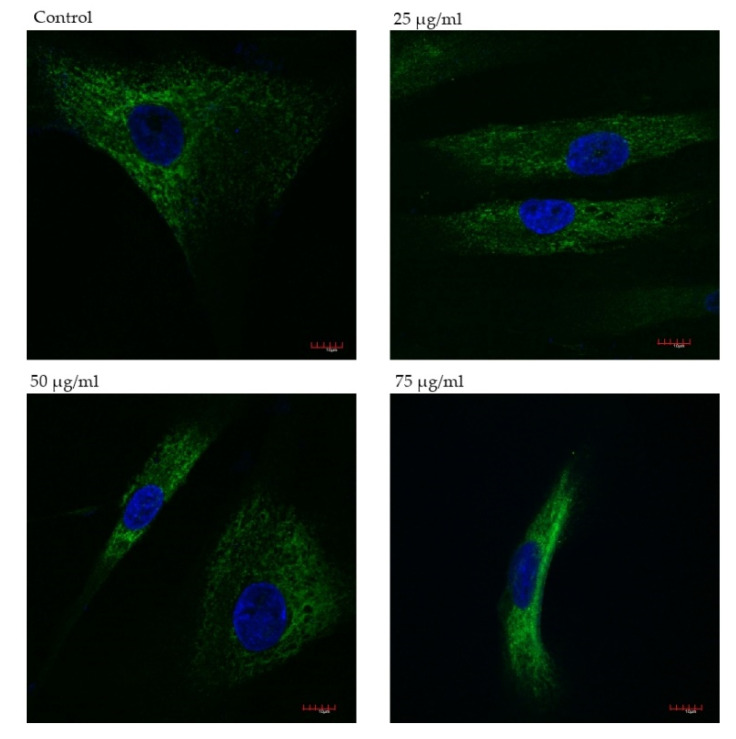
Immunofluorescence staining of collagen type III in fibroblasts after 48 h incubation with salvianolic acid B. Scale bars represent 100 µm.

**Figure 5 ijms-22-07728-f005:**
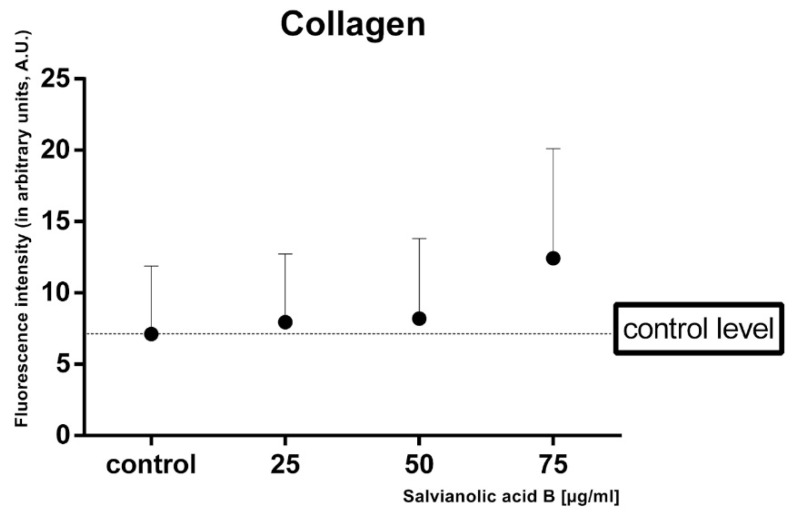
Intensity of fluorescence of collagen type III in fibroblasts in arbitrary units.

**Table 1 ijms-22-07728-t001:** Wound (gap) closure percentage. Data obtained in ImageJ.

Salvianolic Acid B Concentration	Incubation Time		
	24 h	48 h	72 h
Control	20.41%	92.35%	99.40%
25 µg/ml	56.89%	94.72%	99.49%
50 µg/ml	51.63%	83.09%	98.42%
75 µg/ml	16.72%	26.50%	42.09%

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
