# Peer review of "Evaluation of Wound Healing Activity of Salvianolic Acid B on In Vitro Experimental Model"

_ijms, 2021, doi:10.3390/ijms22147728_

Round 1

Reviewer 1 Report

The manuscript entitled “Evaluation of wound healing activity of salvianolic acid B on in vitro experimental model” is dedicated to investigation the influence of salvianolic acid B which is the representative component of phenolic acids derived from the dried root and rhizome of Salvia miltiorrhiza on proliferation, migration, and production of collagen type III by fibroblasts. Authors have shown the high potential of aforementioned acid in wound healing. Salvianolic acid B was found to stimulate the cell migration along with their viability. Generally, the Manuscript is original and has a significance for the scientific community. Experimental and theoretical approach to the discussed problem is presented at a high level. As a reviewer, I have no significant remarks to scientific character of the article.     

In order to improve the manuscript, the following suggestions should be taken into account by the authors:

  1. Please, delete period at the end of Manuscript title.
  2. Page 7, line 202: in vitro instead of in vitro.
  3. Conclusion section should be added to the Manuscript.

After this minor revision, I highly recommend the present manuscript for publication.  

Author Response

Dear Reviewer,

We sincerely appreciate your significant contribution in the revision of our manuscript and giving us another opportunity to improve our manuscript in the best possible way. According to your suggestion, we have improved the manuscript. We are very grateful for appreciated review process.

Reviewer 2 Report

According to my opinion, the manuscript entitled "Evaluation of wound healing activity of salvianolic acid B on in vitro experimental model", given by Urszula Szwedowicz, Anna Szewczyk, Krzysztof Gołąb and Anna Choromańska merits to be published in International Journal of Molecular Sciences. I have no decisive critical remarks concerning this manuscript, which is really interesting and contains a lot of new scientific elements. I only expect to add information referring to the purity of the added Salvianolic acid B, contamination type and amount, if it is possible. To sum up, I propose to accept this manuscript in the current form, after including the mentioned information about purity. I agree with the Authors, that the results obtained in the conducted studies and previous scientific reports on the antibacterial properties and low toxicity of Sal B indicate its high potential in wound healing. Moreover, I agree that their promising results would be a great introduction to in vivo experiments and clinical tests.

Author Response

Dear Reviewer,

We sincerely appreciate your substantial impact on our contribution and giving us an opportunity to improve our manuscript in the best possible way. According to your suggestions, we have improved the manuscript.